# Stochastic Segmentation Networks: Modelling Spatially Correlated Aleatoric Uncertainty

**Miguel Monteiro**
Imperial College London
mm6818@ic.ac.uk

**Loïc Le Folgoc**
Imperial College London
llefolgo@ic.ac.uk

**Daniel Coelho de Castro**
Imperial College London
dc315@ic.ac.uk

**Nick Pawlowski**
Imperial College London
np716@ic.ac.uk

**Bernardo Marques**
Imperial College London
bgmarque@ic.ac.uk

**Konstantinos Kamnitsas**
Imperial College London
kk2412@ic.ac.uk

**Mark van der Wilk**
Imperial College London
m.vdwilk@ic.ac.uk

**Ben Glocker**
Imperial College London
b.glocker@ic.ac.uk

## Abstract

In image segmentation, there is often more than one plausible solution for a given input. In medical imaging, for example, experts will often disagree about the exact location of object boundaries. Estimating this inherent uncertainty and predicting multiple plausible hypotheses is of great interest in many applications, yet this ability is lacking in most current deep learning methods. In this paper, we introduce stochastic segmentation networks (SSNs), an efficient probabilistic method for modelling aleatoric uncertainty with any image segmentation network architecture. In contrast to approaches that produce pixel-wise estimates, SSNs model joint distributions over entire label maps and thus can generate multiple spatially coherent hypotheses for a single image. By using a low-rank multivariate normal distribution over the logit space to model the probability of the label map given the image, we obtain a spatially consistent probability distribution that can be efficiently computed by a neural network without any changes to the underlying architecture. We tested our method on the segmentation of real-world medical data, including lung nodules in 2D CT and brain tumours in 3D multimodal MRI scans. SSNs outperform state-of-the-art for modelling correlated uncertainty in ambiguous images while being much simpler, more flexible, and more efficient.[1]

## 1 Introduction

The task of semantic image segmentation is a highly structured prediction problem where the output label maps should capture the spatial consistency of the objects to be segmented. While casting image segmentation as a dense pixel-wise classification task is at the heart of most machine learning approaches [1–3], this paradigm largely ignores the underlying spatial structure. Methods will often rely on inductive biases to capture structure as opposed to modelling it directly. While this approach may yield reasonable, single deterministic predictions, it is insufficient to model the underlying distribution over multiple plausible outputs. In image segmentation, there is often more than one plausible solution for a given input. The exact location of object boundaries is often ambiguous, and ideally, the model should be able to capture this inherent uncertainty.

Uncertainty can be decomposed into aleatoric, which is inherent to the observations, and epistemic uncertainty, which relates to the ambiguity about the model's parameters and can be explained away with more data [4], e.g., a noisy regression problem with many data points has low epistemic but high aleatoric uncertainty. In segmentation, aleatoric uncertainty is both spatially correlated and heteroscedastic, since an image can have both regions with higher and lower uncertainty. The ideal model should represent the joint probability distribution of the labels at every pixel given the image, enabling sampling multiple plausible label maps.

Because aleatoric uncertainty cannot be reduced by acquiring more data, modelling it explicitly is crucial in risk-sensitive applications. In medical imaging, the images are often noisy, and the boundaries between tissue types may not be well defined, which leads to disagreement even between experts. The ability to automatically generate multiple plausible hypotheses to choose from is of high value in applications such as radiotherapy, where trade-offs have to be made about which anatomical regions to include for invasive treatment. Additionally, providing confidence intervals alongside tumour boundaries would allow uncertainty to be taken into account when making critical decisions.

Fully convolutional neural networks (FCNNs) are the state-of-the-art for semantic segmentation [3, 5, 6]. In principle, FCNNs are probabilistic models, since their output is a set of independent categorical distributions per pixel, parameterised by a softmax layer. Because these distributions are independent given the last layer's activations, sampling from this model would result in spatially incoherent segmentations (grainy label noise in the uncertain regions). We argue that any method that only produces independent pixel-wise uncertainty estimates is unable to generate spatially coherent label maps, and thus incapable of fully capturing the structured uncertainty.

Recent work extends FCNNs to model the joint distribution over labels given the image, allowing for multiple plausible segmentations [7–9]. These methods have rigid, hierarchical, memory-intensive architectures, loss functions with manually tuned hyper-parameters, and require one partial forward pass per new sample. In this paper, we introduce stochastic segmentation networks (SSNs), a lightweight and flexible alternative that efficiently captures correlations between pixels by modelling the logit map as a low-rank multivariate normal distribution. In contrast with previous approaches, our method is less complex, achieves higher predictive performance and can generate multiple samples from a single forward pass. In addition, it can be used with any existing architecture, and its efficiency makes it applicable to high-dimensional problems such as 3D imaging.

## 2   Related work

In data constrained scenarios, Bayesian methods are useful for quantifying epistemic uncertainty for previously unseen examples. Seminal works by MacKay [10] and Neal [11] inspired inference methods in Bayesian deep learning such as Markov chain Monte-Carlo [12, 13] and variational inference methods [14, 15]. These methods focus on estimating the posterior over the weights of a neural network which allows for estimating epistemic uncertainty independently of the task. Ensemble [16] and multi-head [17–19] methods follow a frequentist approach to modelling the weight distributions. In the case of label disagreement or noise, defined as aleatoric uncertainty, the issue is not the lack of data. Still, both uncertainties are complementary. In classification, there is work on estimating aleatoric uncertainty by predicting Dirichlet distributions [20–22] as well as post-training calibration of the predicted class probabilities [23, 24]. In segmentation, attempts at quantifying aleatoric uncertainty on a pixel-wise level [4, 25–27] ignore the joint distribution over labels.

Historically, probabilistic graphical models (PGMs) such as conditional random fields (CRFs) [28, 29] have been used to explicitly model the joint probability distribution over labels. However, the inference was mostly limited to predicting the maximum *a posteriori* (MAP) estimate. Although there is work on obtaining the M-best diverse solutions for a given input image [30, 31], these models are restricted to a fixed number of solutions and have computationally expensive inference. Work on combining PGMs and FCNNs to enforce label dependencies as a post-processing step [6, 32, 33] or even within a single model [34] suffers from the same limitations as classic PGMs when quantifying aleatoric uncertainty.

Recently, Kohl et al. [7, 8] and Baumgartner et al. [9] have built on conditional variational auto-encoders [35, 36] to extend FCNNs for modelling spatially correlated aleatoric uncertainty. Hu et al. [37] extend this framework by regressing the uncertainty maps in a supervised manner. Zhu et al. [38] also make use of deep generative models for the related task of image-to-image translation

with multiple possible outputs for a single input. These methods encode the image into one or more uncorrelated multivariate normal latent variables and rely on the decoder to translate the added uncorrelated stochasticity into meaningful spatial variation. Like variational auto-encoders, these models have the flexibility to transform the latent distributions into arbitrarily complex distributions with correlations between pixels. However, the placement of the latent variables within the network means that one partial forward pass is required for every new sample. Furthermore, this flexibility comes at the cost of having to use a cumbersome variational inference framework which makes use of a training-only posterior network and manually tuned hyper-parameters weighing the Kullback–Leibler divergence regularisation term of the loss. These overly expressive distributions might not justify their cost, a more constrained distribution could suffice and allow the use of simpler inference methods.

## 3  Background

We start by analysing the independence assumptions made to obtain the cross-entropy loss typically used in image segmentation. Consider a standard segmentation problem in which an image, $\boldsymbol{x}$, with $K$ channels and $S$ pixels, maps to a one-hot label map of the same size, $\boldsymbol{y}$, with $C$ classes: $\boldsymbol{x}_i \in \mathbb{R}^K$ and $\boldsymbol{y}_i \in \{0,1\}^C$ for $i \in \{1,\ldots,S\}$. In a classic CNN, the probability of one label, $p(\boldsymbol{y}_i|\boldsymbol{x})$, is the output of a softmax layer taking as input the logit, $\boldsymbol{\eta}_i$. Before any independence assumptions, the MAP estimate for the negative log-likelihood can be written as:

$$- \log p(\boldsymbol{y}|\boldsymbol{x}) = -\log \int p(\boldsymbol{y}|\boldsymbol{\eta})p_\phi(\boldsymbol{\eta}|\boldsymbol{x})d\boldsymbol{\eta}\,, \tag{1}$$

where $p_\phi(\boldsymbol{\eta}|\boldsymbol{x})$ is the probability of the logit map given the image under a model with parameters $\phi$. To obtain the standard cross-entropy loss, we assume that the logit map is given by a deterministic function, $\boldsymbol{\eta} = f_\phi(\boldsymbol{x})$, which means $p_\phi(\boldsymbol{\eta}|\boldsymbol{x})$ can be written as:

$$p_\phi(\boldsymbol{\eta}|\boldsymbol{x}) = \delta_{f_\phi(\boldsymbol{x})}(\boldsymbol{\eta}) = \prod_{i=1}^{S} \delta_{[f_\phi(\boldsymbol{x})]_i}(\boldsymbol{\eta}_i)\,. \tag{2}$$

Due to this deterministic function, given the image and model, the logits, $\boldsymbol{\eta_i}$ for $i \in \{1,\ldots,S\}$, are conditionally independent of each other, i.e., given the image and model, no new information can be gained about a single logit by observing its neighbours. Secondly, we must assume that the labels, $\boldsymbol{y}_i$ for $i \in \{1,\ldots,S\}$, are independent of each other when given their respective logit:

$$p(\boldsymbol{y}|\boldsymbol{\eta}) = \prod_{i=1}^{S} p(\boldsymbol{y}_i|\boldsymbol{\eta}) = \prod_{i=1}^{S} p(\boldsymbol{y}_i|\boldsymbol{\eta}_i)\,. \tag{3}$$

This is a two-part assumption: first, it assumes that labels, $\boldsymbol{y}_i$, are independent of each other when given the full logit map, $\boldsymbol{\eta}$, and second, it assumes that each label, $\boldsymbol{y}_i$, only depends on its respective logit, $\boldsymbol{\eta}_i$, i.e., no new information can be gained about a label by observing the true values of its neighbours. Incorporating the assumptions of equations 2 and 3 into equation 1, and substituting $p(\boldsymbol{y}_i|\boldsymbol{\eta}_i)$ by a categorical distribution parameterised by the softmax transform of $\boldsymbol{\eta}_i$, we arrive at the familiar form for the cross-entropy:

$$- \log \prod_{i=1}^{S} p(\boldsymbol{y}_i|\boldsymbol{\eta}_i) = -\log \prod_{i=1}^{S} \prod_{c=1}^{C} (\mathrm{softmax}(\boldsymbol{\eta}_i)_c)^{y_{ic}} = -\sum_{i=1}^{S} \sum_{c=1}^{C} y_{ic} \log \mathrm{softmax}(\boldsymbol{\eta_i})_c\,. \tag{4}$$

Whereas in image-level classification these independence assumptions may be valid, in segmentation the labels at each pixel are clearly correlated, which should be taken into account.

## 4  Stochastic segmentation networks

In this work, we propose using weaker independence assumptions by using a more expressive distribution over logits. Specifically, we use a multivariate normal distribution whose parameters are the output of a neural network $\boldsymbol{\eta}|\boldsymbol{x} \sim \mathcal{N}(\boldsymbol{\mu}(\boldsymbol{x}), \boldsymbol{\Sigma}(\boldsymbol{x}))$, where $\boldsymbol{\mu}(\boldsymbol{x}) \in \mathbb{R}^{S \times C}$ and $\boldsymbol{\Sigma}(\boldsymbol{x}) \in \mathbb{R}^{(S \times C)^2}$. A non-diagonal multivariate normal distribution is the simplest distribution that models dependencies between pixels. However, the size of the full covariance matrix scales with the square of the number

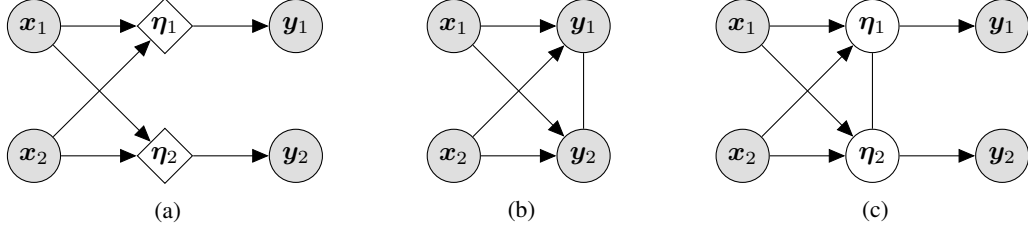

Figure 1: Probabilistic graphical model for a two-pixel segmentation problem: (a) neural network; (b) conditional random field; (c) proposed. $\boldsymbol{x}$ is the image, $\boldsymbol{y}$ the label map and $\boldsymbol{\eta}$ the logits. Circles represent random variables and rhombi represent deterministic variables. Shaded variables are observed and unshaded variables are unobserved.

of pixels times the number of classes making it infeasible to compute for anything but very small images. For this reason, we use a low-rank parameterisation of the covariance matrix of the form:

$$\boldsymbol{\Sigma} = \boldsymbol{P}\boldsymbol{P}^T + \boldsymbol{D}, \tag{5}$$

where the covariance factor, $\boldsymbol{P}$, is a matrix of size $(S \times C) \times R$, where $R$ is a hyper-parameter defining the rank of the parameterisation, and $\boldsymbol{D}$ is a diagonal matrix whose diagonal has $S \times C$ elements. Note that the covariance matrix dependencies are not only spatial but also class-wise. This low-rank parameterisation ensures that the three components describing the distribution: the mean, covariance factor, and covariance diagonal can be efficiently computed by a neural network.

By plugging this distribution into equation 1, we no longer assume that logits, $\boldsymbol{\eta}_i$, are independent of each other. However, the integral also becomes intractable because of the softmax transform on the normal distribution. For this reason, we approximate the integral using Monte-Carlo integration:

$$-\log \int p(\boldsymbol{y}|\boldsymbol{\eta})p(\boldsymbol{\eta}|\boldsymbol{x})d\boldsymbol{\eta} \approx -\log \frac{1}{M} \sum_{m=1}^{M} p(\boldsymbol{y}|\boldsymbol{\eta}^{(m)}), \quad \boldsymbol{\eta}^{(m)}|\boldsymbol{x} \sim \mathcal{N}(\boldsymbol{\mu}(\boldsymbol{x}), \boldsymbol{\Sigma}(\boldsymbol{x})), \tag{6}$$

where $M$ is the number of Monte-Carlo samples used to approximate the integral. Because the distribution only has a few degrees of freedom, the Monte-Carlo integral has low variance. Making use of the assumptions in equation 3 and using the logsumexp operator for numerical stability, we obtain a loss function which we can back-propagate through using the re-parameterisation trick:

$$-\text{logsumexp}_{m=1}^{M}\left(\sum_{i=1}^{S} \log(p(\boldsymbol{y}_i|\boldsymbol{\eta}_i^{(m)}))\right) + \log(M), \quad \boldsymbol{\eta}^{(m)}|\boldsymbol{x} \sim \mathcal{N}(\boldsymbol{\mu}(\boldsymbol{x}), \boldsymbol{\Sigma}(\boldsymbol{x})), \tag{7}$$

where $\log(p(\boldsymbol{y}_i|\boldsymbol{\eta}_i^{(m)}))$ can be solved as in equation 4. For inference, with a single forward pass, we can sample from the distribution multiple times to obtain logit maps, which can be transformed into a probability or label maps. To obtain the most likely logit sample, we use the mean of the distribution.

Figure 1 shows the probabilistic graphical models of a classic neural network, a CRF and the proposed model. While the neural network does not model dependencies between output labels, the CRF explicitly models these dependencies at the cost of an expensive inference procedure. In contrast, by implicitly modelling label dependencies in the logit space and then making independence assumptions, we can capture label dependencies while keeping the efficient inference of a neural network. The overhead of the proposed method is minimal: it involves using three maps instead of one at the end of the network, and sampling from a low-rank normal distribution to compute the loss, which is linear with the rank: $\mathcal{O}(rank)$. Thus, the overall cost is largely dominated by the underlying network.

## 5 Experiments and Results

### 5.1 Toy problem

Consider a dataset on a one-dimensional 21-pixel line with one image for which there are two equiprobable label maps. For both label maps, the first third of the line is labelled 1 (on), and the last third is labelled 0 (off). However, the middle third is off for the first label map, and on for the second

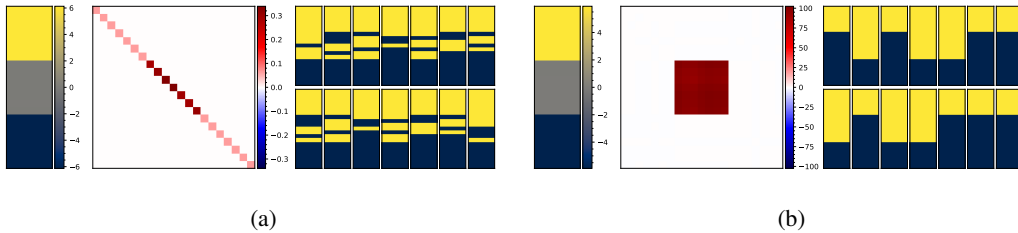

<div align="center">(a)                                        (b)</div>

Figure 2: Toy problem results for the diagonal model (a) and a low-rank model (b). For both sub-figures, from left to right: mean, covariance matrix, and 14 random samples. The mean and samples are one-dimensional but expanded horizontally for improved visualisation. Colour bars indicate intensity values.

label map (see visual examples on the far right of Figure 2). In this setting, the labels of the middle third are uncertain but not independent. Since there is only one input, it is a constant and hence can be disregarded for further modelling. Thus, the goal of the problem becomes to find a generative model for the distribution of the two label maps.

A deterministic model would correctly learn the mean of the distribution but would yield implausible predictions. The first and last thirds would be correct, but the middle third would be arbitrarily fixed. For example, if the label maps were not equiprobable, the model would always generate the most probable one. Next, we consider two stochastic models where the distribution over logits is a multivariate normal distribution: one with a diagonal covariance matrix and one with a low-rank covariance matrix ($\mathrm{rank} = 2$). We train these models with gradient descent and the loss function in equation 7 using 200 Monte-Carlo samples for 10000 iterations. The results are shown in Figure 2. We observe that the diagonal model is able to learn the mean of the distribution and even which pixels have higher uncertainty. However, it cannot learn the structure of the noise and thus produces samples with uncorrelated noise. In contrast, the low-rank model is able to learn the correct noise structure and produce samples matching the desired distribution, yielding a higher log-likelihood, -0.93, when compared to the diagonal model, -4.87.

**Caveat:** Under our model, we can deduce that the true generative model is as follows: the mean is zero for the middle third, $+\infty$ for the first third, and $-\infty$ for the last the third. The covariance matrix is $+\infty$ for all entries regarding self and cross covariances of pixels in the middle third and zero elsewhere. This area of infinite covariance caused numerical stability issues since the covariance quickly grew to infinity producing overflow errors. Furthermore, we found that the covariance grew much faster than the mean causing the model to get stuck in suboptimal local minima. To address these issues, we pre-train the mean first and use early stopping to obtain the last model before an overflow error occurs. In the real data used in this paper, the only area with infinite covariance is the air in the background of brain scans. We addressed the issue by masking out the background.

## 5.2   Lung nodule segmentation in 2D

To compare with previous work, we evaluated our model on the LIDC-IDRI dataset [39] using the task defined by Kohl et al. [7]. The dataset consists of 1018 3D thorax CT scans where four radiologists have annotated multiple lung nodules in each scan. The dataset was annotated by 12 radiologists, and it is not possible to match an annotation to an expert. Thus, the four sets of annotations are not self-consistent in "style" across images. Regardless, this type of data is ideal for validating models which seek to capture the inherent uncertainty in the data — evident from the disagreement between experts. Kohl et al. [7] preprocessed the data by extracting 2D slices centred around the annotated nodules. When at least one expert has segmented a nodule, a slice of the image and four expert segmentations were extracted. Empty segmentations were introduced when there were less than four annotations for a slice. This process yielded a dataset of 15096 slices each having four segmentations.

We compared with three baseline models: a deterministic U-Net [5], a probabilistic U-Net [7] and the PHiSeg model [9] (the best performing variant reported). We used the pre-processed data provided by Kohl et al. [7] and the code, configurations, and hyper-parameters provided by Baumgartner et al. [9], see appendix A.1 for more details on the training procedure. We implemented our algorithm on

Table 1: Quantitative results on the LIDC-IDRI dataset for the five models trained on one set and four sets of annotations. Numbers are presented as mean ± standard error. Arrows in the column headers indicate the direction of increased performance.

| model | trained on | $DSC$ (%) ↑ | $DSC_{nod}$ (%) ↑ | $D^2_{GED}$ ↓ | sample diversity |
|---|---|---|---|---|---|
| deterministic U-Net | set 0 | 37.5 ± 0.4 | 50.3 ± 0.4 | 0.698 ± 0.009 | 0.000 ± 0.000 |
| probabilistic U-Net | | 38.4 ± 0.4 | 57.2 ± 0.4 | 0.516 ± 0.007 | 0.290 ± 0.004 |
| PHiSeg | | 39.1 ± 0.4 | 51.3 ± 0.5 | 0.456 ± 0.008 | 0.215 ± 0.003 |
| proposed (diagonal) | | 37.1 ± 0.4 | 51.2 ± 0.4 | 0.734 ± 0.009 | 0.001 ± 0.000 |
| proposed (low-rank) | | 40.7 ± 0.4 | 58.6 ± 0.4 | 0.365 ± 0.005 | 0.399 ± 0.004 |
| deterministic U-Net | all sets | 35.9 ± 0.4 | 43.5 ± 0.5 | 0.607 ± 0.009 | 0.000 ± 0.000 |
| probabilistic U-Net | | 39.0 ± 0.4 | 50.6 ± 0.5 | 0.252 ± 0.004 | 0.469 ± 0.003 |
| PHiSeg | | 33.8 ± 0.4 | 40.3 ± 0.5 | 0.224 ± 0.004 | 0.496 ± 0.003 |
| proposed (diagonal) | | 37.0 ± 0.4 | 46.2 ± 0.5 | 0.622 ± 0.009 | 0.007 ± 0.001 |
| proposed (low-rank) | | 43.6 ± 0.4 | 68.5 ± 0.3 | 0.225 ± 0.002 | 0.609 ± 0.002 |

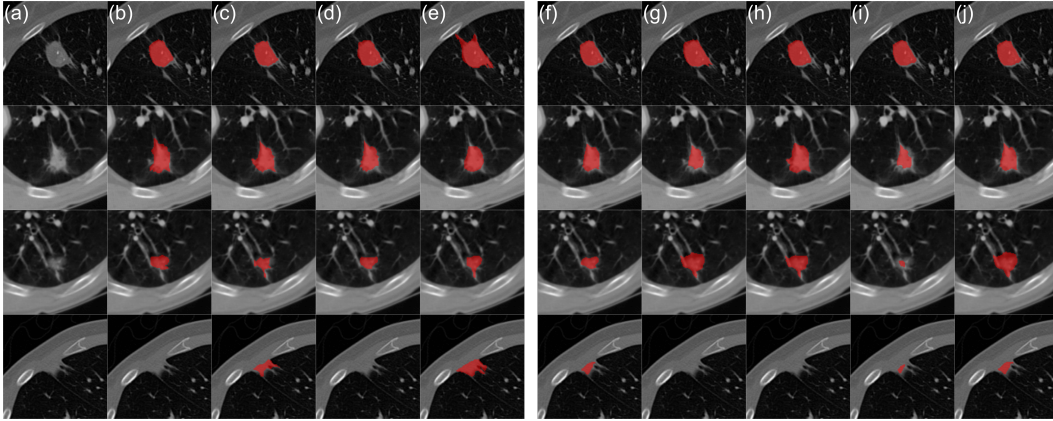

Figure 3: Qualitative results on the LIDC-IDRI dataset for the proposed model trained on four expert annotations: (a) CT image; (b-e) radiologist segmentations; (f) mean prediction; (g-j) samples.

top of the provided deterministic U-Net with $\text{rank} = 10$ for the low-rank model, and, for comparison, we tested a model with a diagonal covariance matrix. By using the same backbone, code and hyper-parameters, we ensured a fair comparison with previous work.

We measured the predictive performance using the Dice Similarity Coefficient, $DSC = \frac{2TP}{2TP+FN+FP}$, where $TP$ is true positives, $FN$ is false negatives, and $FP$ is false positives. Even if all four radiologists annotated a nodule, disagreements about its borders combined with the 3D to 2D preprocessing introduce several empty annotations (on average 1.6/4 = 40.4%). A non-empty prediction on an empty annotation results in a zero towards the average $DSC$, heavily penalising it. Therefore, we also report $DSC_{nod}$ defined as the $DSC$ computed only where the ground-truth annotations are not empty. Pixel-wise metrics for uncertainty quantification and calibration are not appropriate for spatially structured prediction such as segmentation. Hence, we used sample diversity to quantify the amount of uncertainty, and the distance between the expert and predicted distributions to quantify uncertainty calibration. Given the ground-truth distribution defined by the four expert segmentations, $p$, and the predicted distribution, $\hat{p}$, we measure the distance between the two using the generalised energy distance [7, 40]:

$$D^2_{\text{GED}}(p, \hat{p}) = 2\,\mathbb{E}_{y \sim p, \hat{y} \sim \hat{p}}[d(y, \hat{y})] - \mathbb{E}_{y, y' \sim p}[d(y, y')] - \mathbb{E}_{\hat{y}, \hat{y}' \sim \hat{p}}[d(\hat{y}, \hat{y}')], \qquad (8)$$

where $d = 1 - \text{IoU}(\cdot, \cdot)$, if both segmentations are empty $d = 0$. We define sample diversity as $\mathbb{E}_{\hat{y}, \hat{y}' \sim \hat{p}}[d(\hat{y}, \hat{y}')]$. Note how both these metrics are bounded between zero and one.

To measure how models deal with increasing uncertainty in the labels, we trained each model using only one and all four annotations per image. We divided the data into train, validation and test

Table 2: Quantitative results on the BraTS 2017 dataset. Numbers are presented as mean $\pm$ standard error. Arrows in the column headers indicate the direction of increased performance.

| model | $DSC_{WT}$ (%)$\uparrow$ | $DSC_{NET}$ (%)$\uparrow$ | $DSC_{OD}$ (%)$\uparrow$ | $DSC_{ET}$ (%)$\uparrow$ | $D^2_{GED}\downarrow$ | sample diversity |
|---|---|---|---|---|---|---|
| Deepmedic | $88.2 \pm 1.3$ | $60.5 \pm 2.9$ | $72.1 \pm 2.3$ | $67.3 \pm 3.5$ | $0.886 \pm 0.043$ | $0.000 \pm 0.000$ |
| low-rank 30 mm | $88.0 \pm 1.3$ | $59.3 \pm 3.1$ | $71.7 \pm 2.3$ | $68.7 \pm 3.5$ | $0.635 \pm 0.029$ | $0.312 \pm 0.014$ |
| low-rank 60 mm | $88.7 \pm 1.3$ | $59.6 \pm 3.0$ | $72.4 \pm 2.2$ | $69.2 \pm 3.5$ | $0.689 \pm 0.031$ | $0.217 \pm 0.012$ |

sets (60/20/20%), and trained all models for 500k iterations with the same configuration described in Baumgartner et al. [9]. For the proposed loss function, we used 20 Monte-Carlo samples. We computed $D^2_{GED}$ and sample diversity using 100 random samples. The prediction for the probabilistic baselines was obtained by averaging the probability maps of these samples [9]. For the proposed model, we used the mean of the logit map distribution. We computed the $DSC$ between the prediction and the four ground-truths before averaging over sets of annotations and slices.

Table 1 shows the results for the five models and Figure 3 shows qualitative results for the proposed low-rank model trained on four sets of annotations. In terms of predictive performance, the proposed low-rank model outperformed the baselines for both settings.[2] Of note, our model is the only method which benefits from the additional annotations yielding improved predictive performance. For uncertainty calibration, our model yielded the lowest $D^2_{GED}$ except for the PHiSeg model with four annotations where their performance was comparable. In both settings, the proposed and baseline models obtained some measure of sample diversity, while the diagonal model nearly collapsed to a deterministic model, yielding very little sample diversity. For reference, the diversity between experts is $0.399 \pm 0.002$.

## 5.3 Brain tumour segmentation in 3D

We also applied our method to the BraTS 2017 dataset [41–43]. This dataset consists of 285 3D multimodal MRI images (four channels: T1, T1ce, T2 and Flair) where one radiologist has segmented four classes: background, non-enhancing/necrotic tumour core (NET), oedema (OD) and enhancing tumour core (ET). We implemented the proposed method on top of an implementation of DeepMedic [33, 44], a network specifically developed for brain segmentation. We use $\text{rank} = 10$ for the low-rank model and omit the diagonal only model since it converged to a deterministic model. For an ablation study on the impact of the rank on the performance metrics see appendix A.3. The images have a resolution of $1 \times 1 \times 1$ mm and a size of $240 \times 240 \times 155$ voxels, making them too large to train on whole images. We trained the baseline and proposed models on image patches of 110 mm$^3$ (1mm$^3$ = 1 voxel), which, since no padding was used, result in label map patches of 30 mm$^3$. To test the effect of including longer distance dependencies between voxels, we also trained the proposed model on image patches of 140 mm$^3$ which result in label map patches of 60 mm$^3$. Note that, increasing the patch size of the baseline does not change its behaviour since the model is fully convolutional and its receptive field is 81 mm$^3$ (which is larger than 60 mm$^3$).

We split the data into training, validation and test sets (60/10/30%) and trained according to the procedure described in appendix A.1. During inference, we stitched together the patches of the mean, covariance factor and diagonal to build a distribution over the entire image from which we can sample, this ensures no artefacts appear at patch borders. Due to the fully convolutional nature of the model, after it is trained, the patch size used for inference has no impact on the final result. We measured the $DSC$ of the three lesion classes and the whole tumour (WT), consisting of all lesion combined. We measured sample diversity and $D^2_{GED}$ using only 20 samples due to the quadratic dependency on the number of samples and the large image size.

Table 2 shows the quantitative results for the deterministic and stochastic models. The stochastic models had no loss in performance when compared to the deterministic model. Comparing the two stochastic models, we observe that the added spatial context did not increase performance or yield a better-calibrated distribution. Regardless, the amount of needed spatial context is application dependent. Figure 4 shows qualitative results for six cases for the stochastic 30 mm$^3$ model. We observe entire structures in the segmentation appear and disappear between samples in regions of high uncertainty (e.g., row 4). Furthermore, mistakes made by the deterministic model or the stochastic

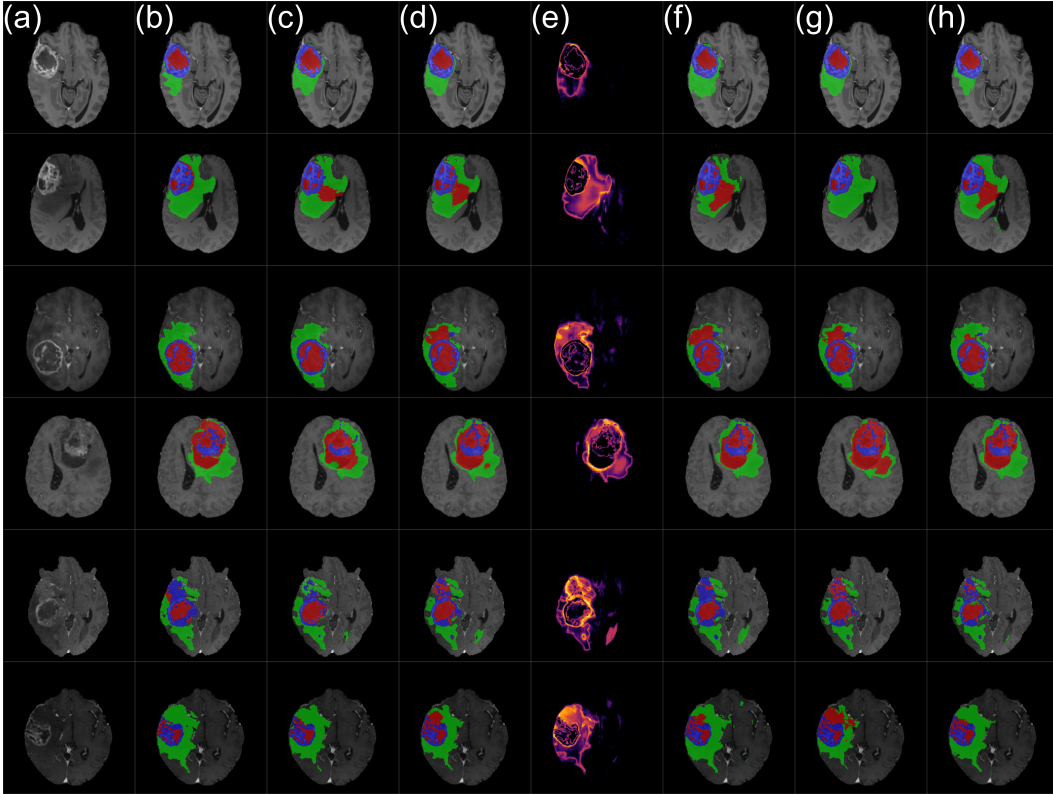

Figure 4: Qualitative results on the BraTS 2017 dataset for 30 mm$^3$ model: (a) T1ce slice; (b) ground-truth segmentation; (c) prediction of deterministic model; (d) prediction of proposed model; (e) marginal entropy; (f-h) samples. Samples were selected to show diversity.

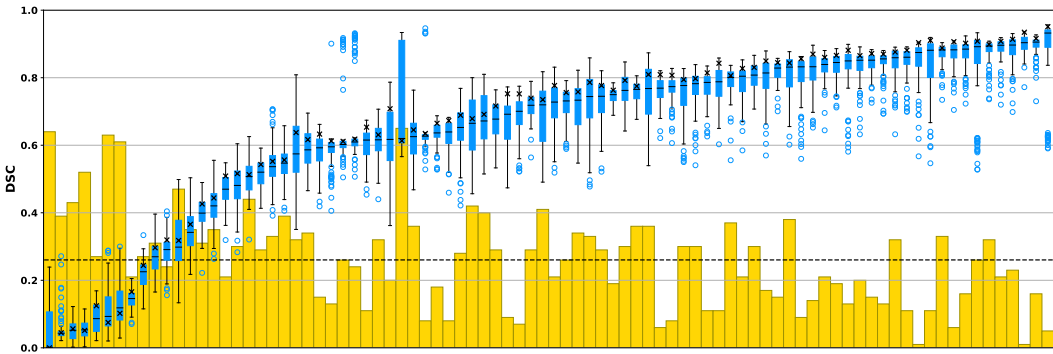

Figure 5: Distribution of sample average class $DSC$ per case. The yellow bars denote the fraction of samples whose $DSC$ is higher than the mean prediction, which is represented by a cross. The dashed line is the average fraction of samples better than the mean prediction (average height of the bars).

model are corrected in at least one of the samples (e.g., row 2). Lastly, the high uncertainty in lesion borders makes them shrink and expand consistently between samples (e.g., row 1). For more samples see appendix A.4.

Figure 5 shows per case sample distributions of the average lesion class $DSC$ (100 samples). As expected, for most cases, the majority of samples are worse than the mean prediction. However, on average, 26.0% samples are better than the mean prediction (dashed line). When looking at the best samples, the average (over the dataset) 95% quantile of the average class $DSC$ was 70.3% when compared with the deterministic model average class $DSC$ of 66.8%. This gain is not uniformly distributed as it tends to be higher for cases with low performance and decrease as the performance

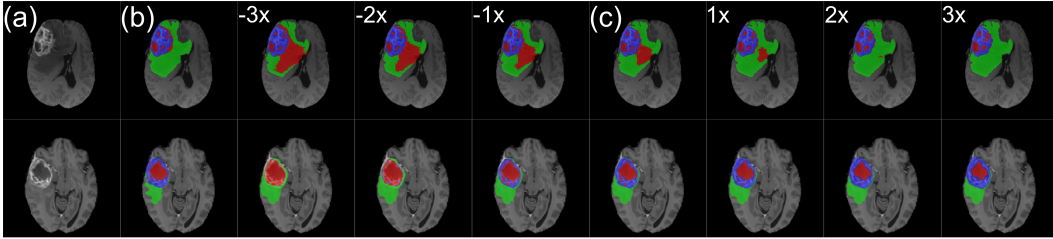

Figure 6: Sample manipulation after inference: (a) T1ce slice; (b) ground-truth segmentation; (c) sample surrounded by manipulated sample with scaling ranging from -3x to 3x.

increases. In addition to being able to sample repeatedly after inference, another advantage of outputting a full distribution is the ability to manipulate samples post-inference. Since the covariance matrix has entries which are separable per class, by scaling only the part of the matrix relating to a given class, we are able to manipulate samples to increase or reduce the presence of that class. This can be used to correct possible mistakes or adjust borders, as shown in Figure 6. Similarly, we can trade sample diversity for quality by scaling the temperature of the entire distribution.

## 6    Discussion

This paper introduces an efficient approach for modelling spatially correlated aleatoric uncertainty in segmentation. We have shown that our method outperforms the baselines while being much simpler, improves predictive performance with added uncertainty, and the samples it generates can be better than those of a deterministic approach. The simplicity of the method enables it to be easily implemented over any existing neural network architecture, which enabled its use in a 3D application, something which had previously not been attempted. The ability to generate multiple plausible hypotheses post-inference is of value in human-in-the-loop scenarios, such as radiology, where a human could manipulate the segmentation semi-automatically according to the model's uncertainty. Furthermore, even in fully-autonomous systems such as autonomous vehicles being able to reason about spatially correlated uncertainty is essential. For example, uncertainty about whether a region is a pedestrian or not should be correlated over all pixels in the region.

## Broader Impact

Proper uncertainty quantification is crucial to increase trust and interpretability in deep learning systems, which is of particular importance in healthcare applications. Reliable uncertainty estimates could help inform clinical decision making, and importantly, provide clinicians with feedback on when to ignore automatically derived measurements. Moreover, uncertainty estimates could be propagated to downstream clinical tasks such as radiotherapy planning, e.g., the amount of radiation delivered to each anatomical region. In medicine, the notion of a second opinion is well established and an essential part of scrutinising the decision process. The ability to generate and manipulate multiple plausible hypotheses could be of great benefit in semi-automatic settings, such as machine aided image segmentation, and help minimise the risk of missing important modes of the target distribution. A complementary prediction might be contradictory yet still very informative.

## Acknowledgements

This research has received funding from the European Research Council (ERC) under the European Union's Horizon 2020 research and innovation programme (grant agreement No 757173, project MIRA, ERC-2017-STG). NP is supported by a Microsoft Research PhD Scholarship. DC and NP are also supported by the EPSRC Centre for Doctoral Training in High Performance Embedded and Distributed Systems (HiPEDS, grant ref EP/L016796/1). LLF is funded through the EPSRC (EP/P023509/1).

## Footnotes

[1]Code available at: https://github.com/biomedia-mira/stochastic_segmentation_networks

[2]The $DSC$ reported for the baseline models is different from what is reported in the literature because we calculate $DSC$ differently, see appendix A.2.

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
