[Supplementary Material]

# A Appendix

## A.1 Training procedure for the LIDC and BraTS datasets.

For the LIDC dataset, all methods were trained using the same configuration and hyper-parameters as in [9]. The models were trained for 500000 iterations using the Adam optimiser [45] with a learning rate of 0.001 and a batch size of 12. The images were randomly augmented through flipping, rotating, and scaling.

For the BraTS dataset, we split the data into training, validation and test sets (60/10/30%) and trained the models for 1200 epochs. At each epoch, we randomly sampled 50 images and extracted 20 patches from each image. We randomly sampled the patches centred around a lesion or background voxel with equal probability. We used the RMSProp optimiser [46] with momentum 0.6 and a learning rate of 1e-3 which we halved at the following epochs: 440, 640, 800, 900, 980, 1050. For augmentation, we used random elastic deformations, right-angle rotations, flips and linear intensity transformations. We used a batch size of 10, except for the 60 mm$^3$ model where we used a batch size of 4 due to GPU memory constraints.

## A.2 Evaluation details.

To calculate the distance between two label maps we used $d = 1 - \mathrm{IoU}(\cdot, \cdot)$. To calculate the IoU in a multi-class setting, we averaged over the IoU of the individual classes, excluding the background class. If both label maps are empty $d = 0$. The $DSC$ (which is equivalent to the F1-score) reported in our work is lower than the results reported in PHiSeg [9]. The authors used a convention where the $DSC$ is 1.0 if both the predicted and ground-truth slices are empty. We argue that this choice skews results since an algorithm that always predicts an empty label map would achieve an average $DSC$ equal to the fraction of empty slices in the dataset, e.g. if the dataset has 40% of empty slices the average $DSC$ is also 40%. In contrast, we used the standard definition of $DSC$, where these cases are undefined and thus excluded from the calculation of the average $DSC$. This changes the range of the numbers we report but not the underlying performance. When we calculated the $DSC$ using the convention used in previous literature, we observed the baseline models performance to match that of what was previously reported in [9]. To calculate uncertainty maps, we used the marginal entropy of the categorical distributions predicted for each voxel $i$:

$$H[y_i|\boldsymbol{x}] = \mathbb{E}_{\boldsymbol{x}}\left[ -\sum_{c=1}^{C} p(y_i = c|\boldsymbol{x}) \log_C p(y_i = c|\boldsymbol{x}) \right] \approx \mathbb{E}_{\boldsymbol{x}}\left[ -\sum_{c=1}^{C} \bar{p}_{ic} \log_C \bar{p}_{ic} \right], \quad (9)$$

where $\bar{p}_{ic} = \frac{1}{M}\sum_{m=1}^{M} p(y_i = c|\boldsymbol{\eta}_i^{(m)}) \approx \mathbb{E}_{p(\boldsymbol{\eta}|\boldsymbol{x})}[p(y_i = c|\boldsymbol{\eta}_i)] = p(y_i = c|\boldsymbol{x})$.

## A.3 Rank ablation study

Intuitively, the rank of the multivariate normal distribution controls the number of independent clusters of pixels that are controlled together, thus, limiting the maximum possible sample complexity. In this section, we provide an ablation study of how the rank of the multivariate normal distribution impacts the performance metrics on the BraTS dataset using models trained on 110 mm image patches. Figure A1 shows the sample diversity, generalised energy distance and average class $DSC$ for different six rank values: $rank \in [1, 2, 5, 10, 15, 20]$. The results are shown as the mean and standard error over five random seeds, that is $6 \times 5 = 30$ total training runs. We observe that as long as the rank is greater than one, there seems to be no clear relation between the rank and the performance metrics. From Figure A2, we see that increasing the rank increases the visual sample complexity, with more intricate structures appearing. Even though we haven't quantified sample complexity, we speculate that the increase in sample complexity does not improve performance because the structure of the aleatoric uncertainty in this dataset is very simple. This property is dataset-specific, which should be taken into account when choosing the rank for a new dataset.

## A.4 Extra figures for the BraTS dataset

Figure A3 compares sampling from the independent categorical distributions of a deterministic model with sampling from the proposed model. Notice the grainy label noise for the deterministic model. Figures A4 - A7 show additional random samples for the stochastic model for multiple test cases.

(a) Sample diversity.

(b) Generalised Energy Distance

(c) Average Sample $DSC$

Figure A1: Impact of rank on different performance metrics for the BraTS dataset. Results are shown as mean and standard error over five random seeds.

Figure A2: Visual impact of rank on samples for one case. Each row represents a model with different rank, and each column a different sample. Rank is increasing from top to bottom: $rank \in [1, 2, 5, 10, 15, 20]$.

Figure A3: Sampling from the independent categorical distributions (top) versus the proposed model (bottom). From left to right: T1ce slice; ground-truth; marginal entropy; five random samples.

Figure A4: Results of sampling from the proposed stochastic model. From left to right: T1ce slice; ground-truth; prediction of deterministic model; prediction of stochastic model; marginal entropy; seven random samples.

Figure A5: Results of sampling from the proposed stochastic model. From left to right: T1ce slice; ground-truth; prediction of deterministic model; prediction of stochastic model; marginal entropy; seven random samples.

Figure A6: Results of sampling from the proposed stochastic model. From left to right: T1ce slice; ground-truth; prediction of deterministic model; prediction of stochastic model; marginal entropy; seven random samples.

Figure A7: Results of sampling from the proposed stochastic model. From left to right: T1ce slice; ground-truth; prediction of deterministic model; prediction of stochastic model; marginal entropy; seven random samples.