[Reviews · NeurIPS 2020]

Review 1

Summary and Contributions: In this paper, the authors suggest a simple and efficient stochastic method for modeling the aleatoric uncertainties, by directly modeling the output logits as multivariate Gaussian distribution. Assessment of model performance on three datasets, including a toy problem and two publicly available medical imaging datasets shows improvements in most of the metrics concerning segmentation quality, uncertainty calibration, and sample diversity.

Strengths: * The paper is tackling an important problem, that is of interest to the community; modeling aleatoric uncertainties and producing multiple plausible coherent outputs, which is quite relevant and important n safety-critical applications such as medical image analysis. * The method is simple and can be easily plugged into the existing method. Besides, the computational overheads are seemingly small due to having the sampling process at the end, while numerical results show still reasonable sample diversity. * Empirical results show improved performance in terms of sample diversity, uncertainty calibration, and segmentation quality.

Weaknesses: * The paper can still improve further in its presentation clarity and further interpret the observed results. * The approach is not radically novel in the methodological contributions.

Correctness: I have no major concerns regarding the correctness of the presented materials; there are clarifications and minor corrections needed.

Clarity: The paper is mostly clear and easy to read, though certain improvements are possible. Details will follow in additional feedback.

Relation to Prior Work: The two most relevant works, i.e. from Kohl et al., and Baumgartner et al., are covered. However, I was expecting to also see discussions and comparisons on multimodal image-to-image translation techniques, e.g. from Zhu et al.. Zhu et al., Toward multimodal image-to-image translation. Nuerips 2017.

Reproducibility: Yes

Additional Feedback: Following are my further detailed comments: * The sample diversity metric is more difficult to interpret as it's not the case that necessarily the higher would be the better. It's useful that the authors have also provided the sample diversity metric for the manual annotation set (0.399) as a reference. However, we observe that the proposed low-rank model is reporting a sample diversity (0.609) that is much higher than the diversity of real annotations. An explanation regarding this would be helpful. * The authors mention that: "During inference, we stitched together the patches of the mean, covariance factor and diagonal to build a distribution over the entire image from which we can sample". Am I understanding it correctly that with this the model wouldn't be able to capture the inter-patch correlations? Wouldn't this impose artifacts at the stitched patch borders? * Line 43: reference 5, is not a good reference for fully convolutional semantic segmentation networks and should be removed. * Line 95: I guess the authors meant p(y_i, x) rather than p(y_i, x_i). line 101-102: "the logits \eta_i for i \in {1, ..., S} are trivially independent" => "conditionally independent", I think. * Line 104: "i \in {0, ..., S}" => "i \in {1, ..., S}" * Equation (6): Please introduce M. * Figure 1: The graphical notation of using Rhombus needs a short clarification in the figure caption. * Figure 2: The means representations in the first and fourth leftmost figures are rather confusing given the numbers on their right side (-6 to 6). Wasn't the output 21 dimensional? ========post-rebuttal feedback: I have carefully read the reviews and responses from the authors. I appreciate that the authors have addressed most of my concerns regarding the presentation quality, comparison to the relevant literature on multi-modal image-to-image translation as well as clarifications on some technical details. I share the view with the authors that the simplicity of the proposed method would help broaden the applicability. Overall, I think this is a good contribution, and I have accordingly adjusted the rating.


Review 2

Summary and Contributions: The paper is concerned with producing spatially correlated distributions over semantic segmentations that are cheap to sample from. As such the approach can model segmentation ambiguities in images and produce samples that are spatially coherent and befit the expected distribution of plausible annotation. The proposed method shows improved performance over prior art on a medical segmentation task considered for probabilistic segmentation by the literature and is further applied to an additional dataset of 3d medical images. The method is comparatively simple which allows it to be incorporated into existing segmentation models with relative ease.

Strengths: The method predicts a low-rank decomposition of the full covariance matrix between the logits of all output pixels. Together with a predicted mean for each pixel, they parametrize a multivariate Gaussian capturing spatial correlations in the predicted segmentations. While conceptually simple, this looks sound and appears to work well. The method is benchmarked against 2 baselines for probabilistic segmentation using the multi-grader LIDC dataset that was considered by prior art. Although the reported results appear to stem from a retraining of the baselines (which is less clean than comparing against the reported numbers on the exact same data splits that the baselines were trained on), these results look credible and are supported by a toy experiment and further qualitative results.

Weaknesses: The method itself does not have many obvious limitations. Perhaps one being the reported training instabilities for when image regions show very low entropy / noise / uncertainty, leading to infinite covariances (see toy example and air in the brain scans). A few things regarding the method could be more clearly presented or discussed further. Among them: 1) It is perhaps not entirely clear why the approach shouldn’t experience any mode-collapse. It’s possible that this is because the inner product taken in the covariance decomposition pretty much always gives positive correlations for similar pixels. 2) The mechanism of calculating the covariance matrix could be presented more intuitively, e.g. in terms of the aforementioned inner-product between locations (and labels), thus amounting to a cosine similarity measure between output pixels. 3) It could be stated still more explicitly that the covariance is spatial and not between classes for each logit. The evaluation presented on LIDC does raise some questions. For one, it appears that the baseline of reference 8 and 9 were retrained on the 60-20-20 data splits, but this is not explicitly mentioned. For another it is not clear whether these splits correspond to the same splits used in reference 9 (which are in fact different from the ones used in reference 8). If they are the same, it is a bit curious to see by how much the performance of the re-trained reference 8 and 9 baselines differs from what is reported in reference 9. Additionally the paper omits further literature concurrent to reference 9 (`A Hierarchical Probabilistic U-Net for Modeling Multi-Scale Ambiguities’), which should also be cited here. The statement that prior work requires a full forward pass for each task (line 51 and 84) doesn’t quite hold and should be toned down. For example the reference 8 baseline only requires re-evaluation of a small block of the network for each sample. The presented figures could be improved with further details: 1) Figure 2 is hard to read, more labels would be helpful. E.g. it is not intuitively clear that positions on the line are vertical here. I would also suggest turning left / right into a)/b) with corresponding subfigure labels. 2) Fig. 5 could also use a legend with colors and an x-axis label. 3) Fig. 6: Unclear what was scaled (what class?). With respect to Figure 2 (right), wouldn’t one expect a block diagonal covariance matrix here? On the LIDC task, the model training is said to be carried out with the same configuration as in reference 9, which is not explicit enough. The exact setup should be reported to allow for reproducibility (can be done in the appendix for example). On LIDC: Why was the mean of the logit map used as the prediction of the proposed approach versus for the baselines the samples were averaged? Both the proposed method and the probabilistic models allow to take a predicted mean or samples. To further improve the fit with the NeurIPS community and show additional comparison against baselines non medical image experiments beyond the toy task would have been nice.

Correctness: Looks correct, although some details around the evaluation of the baselines on LIDC are missing / not clear (see above).

Clarity: The paper is well written.

Relation to Prior Work: For the largest part yes. Small corrections and an additional citation should be incorporated (see above).

Reproducibility: Yes

Additional Feedback: Suggestion: Might be nice to explore how the performance changes with the rank of the variance matrix? [Feedback after reading the rebuttal]: I thank the authors for addressing my comments and clarifying the misconception I had on the covariance being applied to spatial and class dimensions jointly. I further appreciate the updates and additions that the authors will make to their paper. One concern that remains is showing the proposed method's performance on a non-medical segmentation task, such as e.g. Cityscapes.


Review 3

Summary and Contributions: The paper proposes a simple way to model a joint distribution over the whole segmentation mask (as opposed to treating each pixel independently) without the hassle of working with CRFs. The proposed technique uses a low-rank Gaussian distribution to model the segmentation masks.

Strengths: The proposed technique is simple and seems easy to implement and apply. It can be built atop of any segmentation network. It can also be easily extended to 3d data. The model clearly improves upon the considered baselines (probabilistic U-net and PHi-Seg), achieving higher sample diversity and more precise and spatially coherent segmentation masks.

Weaknesses: The authors focus on medical imaging to evaluate their model. It would be interesting to see whether the results would hold on other domains as well. Also, it would be nice to see a more thorough investigation of the properties of the model. How does the rank of the Gaussian influence training time, inference time and predictive performance? Would it make sense to train with a higher rank and then reduce the rank post-training to reduce the inference complexity? Would a more elaborate distribution, e.g. a mixture of Gaussians or an implicit probabilistic model, improve the predictive performance even further?

Correctness: The proposed model and training procedure are sound. The empirical evaluation also seems correct.

Clarity: The paper is clearly written and easy to follow.

Relation to Prior Work: The authors position their work against the works on probabilistic graphical models and against existing works on probabilistic/stochastic networks for segmentation. The similarities and differences are sufficiently discussed and to the best of my knowledge, the proposed technique appears novel. However, I do not follow recent literature on segmentation.

Reproducibility: Yes

Additional Feedback: UPDATE: I have read the author's feedback. My concerns have been adequately addressed and I am now more confident in my score. On a side note: It seems quite easy to visualize the predicted covariance matrix when the rank is low. I suspect each column of the matrix P to highlight a patch of the input image where the predictions are expected to correlate. Such pictures could be beneficial to show the structure of the predicted covariance matrix and could support the intuition "Intuitively, the rank controls the number of independent clusters of pixels that are controlled together"


Review 4

Summary and Contributions: A common assumption in image segmentation tasks is that the pixel level label maps are independent. In this work, a weaker independence assumption is used to model and capture the spatially correlated uncertainty in segmentations and proposed as the stochastic segmentation networks (SSNs). A low rank parameterisation of multivariate Gaussian density over the predicted label maps is used to generate multiple feasible segmentations from which aleatoric uncertainty is quantified. Experiments on a synthetic example and two medical imaging datasets demonstrate the usefulness of the proposed method.

Strengths: + Authors have identified a commonly used assumption of independence of pixel level label maps and alleviate it with a weaker independence assumption. This is a simple but elegant contribution. + Using low-rank approximation of covariance matrix and computing the log-likelihood from samples with reparameterisation is a nice approximation trick to overcome the computational expense. + As the authors also point out, this is one of the first models to provide means to estimate multiple samples for high-dimensional data such as 3D image data which are commonly encountered in medical imaging. + Proposed framework can be integrated into existing segmentation frameworks turning them into stochastic ones which could be valuable + Experiments are thorough and show that SSNs are capable of capturing sample diversity in both 2D and 3D data

Weaknesses: - The choice of the specific weak independence assumption is not sufficiently motivated. It makes sense to use a more expressive distribution but motivation on the why multivariate normal and the specific low-rank parameterisation is stated matter-of-factly. Clearer motivation of this in Sec. 4 might be useful - Whenever a free parameter is introduced, such as the rank R, experiments showing how the specific value was arrived at can be insightful. This also allows readers to appreciate the influence of such, I am assuming, critical parameters. - Both 2D and 3D experiments were performed with rank, R=10. What is the influence of this parameter on sample diversity for other values of R? - What is the computation overhead of introducing SSNs on the U-net in 2D data and on DeepMedic in 3D data experiments? If SSNs should be usable with other (already expensive models) the overhead is important to be reported.

Correctness: Claims in the work are substantiated with relevant experimental evaluation.

Clarity: The paper is largely clearly written.

Relation to Prior Work: The discussions on prior art is adequately done and also comparisons with relevant models are reported. However, the criticism on methods like Prob. U-net and PhiSeg that their "overly expressive distributions might not justify their cost" is unsubstantiated. Perhaps a comparison of their complexity or computation time of these models with SSNs could better substantiate this otherwise shallow claim.

Reproducibility: Yes

Additional Feedback: What is the authors' opinion on having more expressive distributions if it were possible to be efficiently computed? For instance, using flow transformations of a base distribution might still allow for efficient computation of the integral. Would this be useful in capturing richer spatial correlation? Or do the authors think the proposed low-rank approximation (with R=10) is adequate? Update after rebuttal: Authors have addressed some of the concerns in my review and have promised to improve the manuscript which is appreciated. I had given a high score for this work and would keep it as it is.

[Author Response · NeurIPS 2020]

We thank the reviewers for the time and effort spent assessing our work. We are grateful for the positive feedback and the suggestions which are helpful to further improve our paper. Please find our responses to requests below.

[**Reviewer 1**] **Presentation clarity can be improved:** Thanks for the detailed suggestions. These will be incorporated. **Not radically novel in the methodological contributions:** We do believe our method is quite novel and makes a strong methodological contribution not found in previous work. Its simplicity is a key strength which will hopefully help with wider adoption. **Discussion of and comparison with multimodal image-to-image translation:** Thank you for the suggestion. We now discuss these techniques in the related work section. **Interpretation of sample diversity metric:** Here, sample diversity is not a metric of quality but an indicator of how different samples are from each other. To measure how closely we match the expert distribution, we present the generalised energy distance. We will clarify this. **Artefacts at the stitched patch borders:** During training, the model is not capturing inter-patch correlations. However, during inference, the distribution is built over the whole image before sampling, preventing artefacts from appearing between patches. If we sample and then stitch (as tried in initial experiments), artefacts would appear.

[**Reviewer 3**] **Mode-collapse:** This can only occur if the covariance becomes zero/negligible, in which case the samples revert to the mean. This can happen even for similar pixels (see toy example). We believe this is a feature, not a bug. The integral of eq 1 and 6 penalises predicting a single mode unless it is 100% accurate, in which case there is no uncertainty. **Intuition about the mechanism of calculating the covariance matrix:** The inner product is not taken on the output pixels but on the covariance factor coming from its separate set of conv-filters. Thus the covariance does not amount to the cosine similarity between output pixels. It would be the cosine similarity between features of the covariance factor if the inner-product was normalised and no diagonal component was added. **Covariance being spatial and not between classes for each logit:** Actually, the covariance is spatial *and* between classes for each logit. We will add some clarification on this. **LIDC evaluation:** The baselines were retrained with new random splits. During development, we contacted the authors of [9], but they were unable to provide their splits. In our experiments, we found [8] and [9] to perform almost equal to the experiments reported in [9]. However, note that in our paper, we report DSC in a different (arguably more correct) manner, see Appendix A.2. We will make this important difference clear in the text. **Literature concurrent to reference 9:** Thank you, we have missed this work and will include it in the updated discussion of related work. **Statement regarding prior work requiring a full forward pass for each task doesn't quite hold:** Thanks for pointing this out. We agree and will change the statement accordingly. **Figures could be improved:** Thanks for the helpful comments on how to improve the figures. **Figure 2 (right), expect a block-diagonal covariance matrix:** The matrix is block-diagonal. However, the first and last blocks have effectively zero variance (label never changes) and hence are not visible. **Missing training details for LIDC:** Thanks for pointing this out, we will include these details in the appendix as suggested. **Mean of the logit map for prediction versus averaging samples:** We kept the baseline experiments as close as possible to the reported state-of-the-art in [9], which used the sample average. For the baselines, the expected value of the output needs to be computed using a sample average due to the neural network in the middle. In our method, the mean of the distribution already represents the expected value of the logit map.

[**Reviewer 4**] **Training with a higher rank and reducing rank post-training:** Thanks, this is a very interesting suggestion that we hadn't considered yet. **More elaborate distributions or an implicit probabilistic model to improve the predictive performance:** While we did not compare with a mixture, what we have shown is that a simple distribution can improve over the implicit probabilistic models (baselines) while having lower complexity. Nevertheless, comparing with a mixture is a good suggestion for further work.

[**Reviewer 5**] **Better motivation for the specific weak independence assumption:** Thanks for pointing this out, we will clarify the motivation behind our choices. The multivariate normal is the simplest distribution that can model correlations between pixels. The low-rank parameterisation is motivated by computational constraints and as a way of controlling the expressiveness of the distribution (see point below).

[**R3**+**R4**+**R5**] **Influence of the rank on predictive performance:** We agree that studying the effect of varying the rank would be insightful. Therefore, we are preparing an ablation study to be added to the appendix showing how performance metrics (including sample diversity) vary with this parameter. Intuitively, the rank controls the number of independent clusters of pixels that are controlled together.

[**R4**+**R5**] **Computational overhead of SSNs and impact of rank on training and inference time:** The computational overhead is minimal. The overall cost is dominated by the forward pass of the underlying network. The overhead is (1) predicting three maps instead of one at very the end of the network (2) Sampling from the low-rank normal distribution to compute the loss. The cost of sampling is linear with the rank ($\mathcal{O}(rank)$). We will add this to the updated paper.

[**R3**+**R4**] **Other application domains:** Medical imaging is among the most critical applications for uncertainty estimation. We hope the methods and results are relevant for the wider NeurIPS community but agree that in future work, other domains could be explored.

[Meta-Review · NeurIPS 2020]

The reviewers have supported the acceptance of this paper after their comments were addressed in the response.